# Energy-Structured Low-Rank Adaptation for Continual Learning

Longhua Li [1 2]  Lei Qi [1 2]  Qi Tian [3]  Xin Geng [1 2]

## Abstract

While orthogonal subspace methods try to mitigate task interference in Continual Learning (CL), they often suffer from energy diffusion across the basis, hindering knowledge compaction and exhausting capacity for future tasks. We observe that output feature drift induced by parameter updates is inherently low-rank, and theoretically prove that preserving parameters along the principal directions of this drift minimizes the output reconstruction error. Motivated by this, we propose **E**nergy-Concentrated and **E**nergy-Ordered **Lo**w-**R**ank **A**daptation ($E^2$-LoRA). By explicitly ordering and concentrating knowledge into leading ranks, $E^2$-LoRA frees capacity for subsequent tasks. Furthermore, we design a dynamic rank allocation strategy to balance stability and plasticity by jointly optimizing energy retention and model plasticity. Extensive experiments across multiple benchmarks demonstrate that $E^2$-LoRA achieves state-of-the-art performance.

## 1. Introduction

Continual Learning (CL) aims to build models that learn new tasks sequentially without forgetting previous knowledge, a capability crucial for real-world AI systems that evolve over time (Rebuffi et al., 2017; Zhou et al., 2023). While pre-trained models (PTMs) have revolutionized CL by providing robust foundational representations (Dosovitskiy, 2020; Zhou et al., 2025a), parameter drift remains a fundamental challenge: sequential fine-tuning causes feature representations to degrade, leading to catastrophic forgetting (French, 1999; Kirkpatrick et al., 2017).

Current parameter-efficient fine-tuning (PEFT) methods (Hu

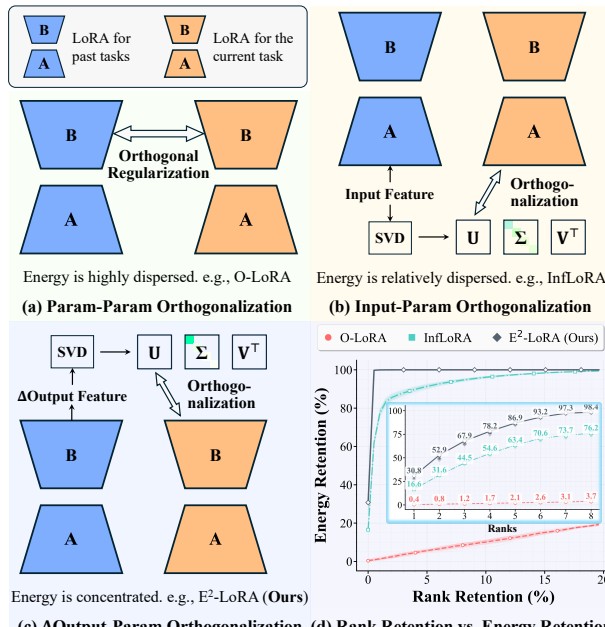

*Figure 1.* Comparison of orthogonal constraints applied in: (a) parameter space; (b) input space; (c) energy-concentrated output drift space (ours). (d) $E^2$-LoRA maintains high energy retention after rank truncation, enabling stable knowledge preservation (for past tasks) and efficient subspace reuse (for new tasks).

et al., 2022; Wang et al., 2022a;b) mitigate this issue by freezing PTM weights and learning lightweight adapters. Orthogonalization-based methods have been proposed to mitigate catastrophic forgetting, and they mainly operate in either the parameter space or the input feature space (Saha et al., 2021; Liang & Li, 2023; Wang et al., 2023; Liang & Li, 2024). However, as illustrated in Figure 1, orthogonalization in both spaces results in energy-dispersed representations, which severely restrict future model plasticity. In contrast, we observe that task-induced output feature drift resides in a far more concentrated and low-dimensional subspace. We further provide theoretical evidence that preserving parameters along the principal directions of this drift minimizes the output reconstruction error. By orthogonalizing LoRA updates within this output-induced spectral space, we can effectively preserve knowledge of past tasks while simultaneously freeing up capacity for learning new ones.

Based on these analysis, we propose **E**nergy-Concentrated

[1]School of Computer Science and Engineering, Southeast University, Nanjing, China [2]Key Laboratory of New Generation Artificial Intelligence Technology and Its Interdisciplinary Applications (Southeast University), Ministry of Education, China [3]Huawei Technologies, Shenzhen, China. Correspondence to: Lei Qi <qilei@seu.edu.cn>, Xin Geng <xgeng@seu.edu.cn>.

*Proceedings of the $43^{rd}$ International Conference on Machine Learning*, Seoul, South Korea. PMLR 306, 2026. Copyright 2026 by the author(s).

and **E**nergy-Ordered **Lo**w-**R**ank **A**daptation ($E^2$-LoRA). For each new task, we allocate a dedicated LoRA module and apply a post-hoc transformation to the learned LoRA such that task-specific knowledge is ordered by rank (*Energy-Ordered*) and concentrated in only a few leading ranks (*Energy-Concentrated*). This design allows the majority of tail ranks to be reinitialized and reused for learning new tasks, thereby preserving the stability of previous tasks while providing sufficient capacity for new ones.

Specifically, after training a LoRA module for a given task, we perform PCA on the output feature shifts induced by the LoRA to obtain an energy-concentrated and energy-ordered basis. By aligning LoRA parameters with this basis, we concentrate task-specific knowledge into leading ranks. When a new task arrives, the parameters corresponding to the low-energy bases of previous tasks are released and repurposed to accommodate learning for the new task. Moreover, by jointly considering the energy retention required for old tasks and the plasticity needed for the new task, we design an adaptive rank allocation strategy that achieves a favorable stability–plasticity trade-off.

Our main contributions are summarized as follows:

- We propose $E^2$-LoRA. Our theoretical analysis shows that $E^2$-LoRA is optimal for preserving knowledge of previous tasks under rank decay, which also facilitates freeing capacity for learning new tasks. Extensive experimental results further validate this property.

- By jointly considering the energy retention of previous tasks and the plasticity required for new tasks, we design an effective strategy for allocating the LoRA rank of each task, achieving a well-balanced trade-off between model stability and plasticity.

- Extensive experiments on both class-incremental learning and domain-incremental learning benchmarks demonstrate that our proposed $E^2$-LoRA consistently outperforms existing methods.

## 2. Related Work

### 2.1. Continual Learning (CL)

Continual Learning (CL) aims to enable models to acquire new classes over time without forgetting previously learned ones (Li & Hoiem, 2017; Rebuffi et al., 2017). A central challenge in CL is catastrophic forgetting, where adapting to new tasks degrades performance on earlier ones (French, 1999; French & Ferrara, 2020), reflecting the fundamental stability–plasticity dilemma. Existing CL approaches can be broadly grouped into several categories. Rehearsal-based methods (Aljundi et al., 2019; Chaudhry et al., 2019; Liu et al., 2020) mitigate forgetting by replaying exemplars from past tasks, but require storing historical data. Knowledge distillation–based methods (Dhar et al., 2019; Douillard et al., 2020; Simon et al., 2021; Tao et al., 2020) transfer knowledge from previous models to the current one, while parameter regularization–based methods (Kirkpatrick et al., 2017; Aljundi et al., 2018) constrain important parameters during updates. Model rectification–based methods (Belouadah & Popescu, 2019; Pham et al., 2022; Shi et al., 2022) adjust inductive biases to reduce incremental prediction bias. More recently, expandable networks (Yan et al., 2021; Wang et al., 2022b; Douillard et al., 2022; Huang et al., 2023; Chen & Chang, 2023; Hu et al., 2023) have achieved strong performance by allocating task-specific backbones and classifiers.

### 2.2. CL with Pre-Trained Models

Large-scale pre-training (Han et al., 2021) has substantially advanced continual learning (CL) by endowing models with strong generalization capabilities. Pre-trained models (PTMs), particularly Vision Transformers (ViTs) (Dosovitskiy, 2020), produce transferable representations that help mitigate catastrophic forgetting. Most PTM-based CL methods freeze backbone weights and rely on parameter-efficient fine-tuning (PEFT) techniques for incremental adaptation (Smith et al., 2023; Yan et al., 2021). Among PEFT approaches, prompt-based methods have received considerable attention. L2P (Wang et al., 2022b), DualPrompt (Wang et al., 2022a), and CODA-Prompt (Smith et al., 2023) adapt PTMs via learnable prompts to encode task-invariant and task-specific knowledge. While effective, these methods remain susceptible to cross-task conflicts and prompt retrieval errors. Adapter-based methods offer an alternative. EASE (Zhou et al., 2024) and MOS (Sun et al., 2025) construct task-specific adapter subspaces to alleviate interference, while TUNA (Wang et al., 2025) integrates task-specific and shared adapters via entropy-based selection and fusion.

More advanced PEFT techniques further improve task separation. O-LoRA (Wang et al., 2023) reduces interference with previous tasks by incorporating an orthogonalization constraint loss between the parameters of prior and subsequent tasks. GPM (Saha et al., 2021) and DualGPM (Liang & Li, 2023) leverage input representations to estimate dominant gradient directions, using them to constrain the gradient updates of subsequent tasks and thereby mitigate forgetting. InfLoRA (Liang & Li, 2024) enforces orthogonality between LoRA modules, and BiLoRA (Zhu et al., 2025) expands the parameter space to approximate orthogonal task subspaces. Recent studies have shown that proxy-based decomposition can transform a parameter matrix into two low-rank matrices while maximizing knowledge retention (Li et al., 2026a;b). Our analysis shows that orthogonalizing subsequent-task parameters against previous-task parameters or input features is suboptimal, as parameter energy is dispersed and input features are high-dimensional

in pre-trained models. However, the task-induced drift in the output features relative to the pre-trained model is low-dimensional and can be effectively captured via SVD decomposition. This allows preserving core knowledge by retaining only dominant directions, freeing parameter space for future tasks while balancing stability and plasticity.

## 3. Methodology

### 3.1. Preliminary and Motivation

Let $\mathbf{W}_0 \in \mathbb{R}^{d_{\text{out}} \times d_{\text{in}}}$ denote the pre-trained weight matrix of a linear transformation (e.g., a projection layer in attention or an MLP). In continual learning, tasks arrive sequentially. Before training on the $t$-th task dataset $\mathcal{D}_t$, the model parameters are denoted as $\mathbf{W}_{t-1}$. After training on task $t$, the parameters are updated as:

$$\mathbf{W}_t = \mathbf{W}_{t-1} + \Delta \mathbf{W}_t, \tag{1}$$

where $\Delta \mathbf{W}_t$ represents the task-specific parameter update.

Recent PEFT methods restrict the weight update $\Delta \mathbf{W}_t$ to a low rank $r$, with $r \ll \min(d_{\text{out}}, d_{\text{in}})$, and fine-tune the model using a LoRA module of rank $r$. The resulting parameter update can be written as $\Delta \mathbf{W}_t = \mathbf{B}_t \mathbf{A}_t$. A key challenge in continual learning is to reduce interference between tasks while preserving sufficient plasticity to learn new tasks. To address this, orthogonalization techniques have been widely adopted, primarily including orthogonalization between the parameters of new and previous tasks (Wang et al., 2023), as well as between the input representations of previous tasks and the parameters of new tasks (Saha et al., 2021; Liang & Li, 2023; 2024).

**Param-Param Orthogonalization.** A straightforward approach (Wang et al., 2023) imposes orthogonality constraints directly on the LoRA parameters $\mathbf{B}_t$ of the current task with respect to those of previous tasks, $\mathbf{B}_1, \cdots, \mathbf{B}_{t-1}$:

$$\mathcal{L}_{\text{orth}}(\mathbf{B}_i, \mathbf{B}_t) = \left\| \mathbf{B}_i^\top \mathbf{B}_t \right\|_F^2. \tag{2}$$

The LoRA parameters of previous tasks are frozen to prevent forgetting. However, since knowledge from earlier tasks is irregularly distributed across the column vectors formed by the $\mathbf{B}$ matrices, and the subspace occupied by old tasks grows linearly as more tasks are added, this strategy severely restricts the learning capacity of new tasks.

**Input-Param Orthogonalization.** Another line of work approximates the dominant gradient directions using input representations (Liang & Li, 2024). After each task is learned, these methods sample input representations and perform SVD to extract their principal directions, then enforce the LoRA parameters $\mathbf{A}_t$ of the new task to be orthogonal to the leading singular vectors of $\mathbf{U}$ of previous tasks' input

representations, thereby effectively reducing interference with earlier tasks. However, in practice, due to the rich representational capacity of pre-trained models, input representations are often highly diverse and energy-dispersed. As a result, constraining parameter updates based on input principal directions can overly restrict the effective update directions, leading to reduced plasticity when learning new tasks. We provide an intuitive analysis in Appendix C.2.

### 3.2. Output Feature Drift–Induced Orthogonalization

In contrast to existing methods, we analyze parameter updates from the perspective of output feature drift. We find that, compared to parameter matrices or input representations, the output drift induced by task-specific LoRA fine-tuning exhibits significantly more concentrated energy. After applying SVD, most of the energy is captured by only a few leading principal directions. By constraining old-task knowledge to these compact directions, this property both preserves existing knowledge and frees up ample, interference-free capacity for learning subsequent tasks.

**Output Feature Drift.** For task $t$, the parameter update $\Delta \mathbf{W}_t = \mathbf{B}_t \mathbf{A}_t$ induces a change in output features:

$$\delta y_t(\boldsymbol{x}) = \Delta \mathbf{W}_t \boldsymbol{x}. \tag{3}$$

Given a batch of input features $\mathbf{X}_t = [\boldsymbol{x}_1, \cdots, \boldsymbol{x}_n]$, the corresponding output drift matrix is

$$\Delta \mathbf{Y}_t = \Delta \mathbf{W}_t \mathbf{X}_t \in \mathbb{R}^{d_{\text{out}} \times n}. \tag{4}$$

Empirically and theoretically, we observe that although input representations $\mathbf{X}_t$ may be high-rank, the induced output drift $\Delta \mathbf{Y}_t$ often lies in a much lower-dimensional subspace, reflecting task-specific semantic changes.

**PCA on Output Drift.** We perform PCA (or equivalently SVD) on the output drift matrix:

$$\Delta \mathbf{Y}_t = \mathbf{U}_t \boldsymbol{\Sigma}_t \mathbf{V}_t^\top, \tag{5}$$

where the columns of $\mathbf{U}_t$ form an orthonormal basis of output directions, ordered by descending singular values (energy). Crucially, we compute this decomposition using a proxy set sampled from the task feature distribution, and apply the resulting singular vectors to perform the energy-based transformation of the parameters $\mathbf{B}_t \mathbf{A}_t$ as described in the following paragraph. This process is conducted after completing training on the current task $t$ and does not require storing any data.

**Energy-Structured Transformation.** We then transform the LoRA parameters accordingly, such that $\mathbf{B}_t$ aligns with the PCA basis $\mathbf{U}_t$:

$$\mathbf{B}_t \leftarrow \mathbf{U}_t[:, : r_t], \quad \mathbf{A}_t \leftarrow (\mathbf{U}_t[:, : r_t])^\top \mathbf{B}_t \mathbf{A}_t, \tag{6}$$

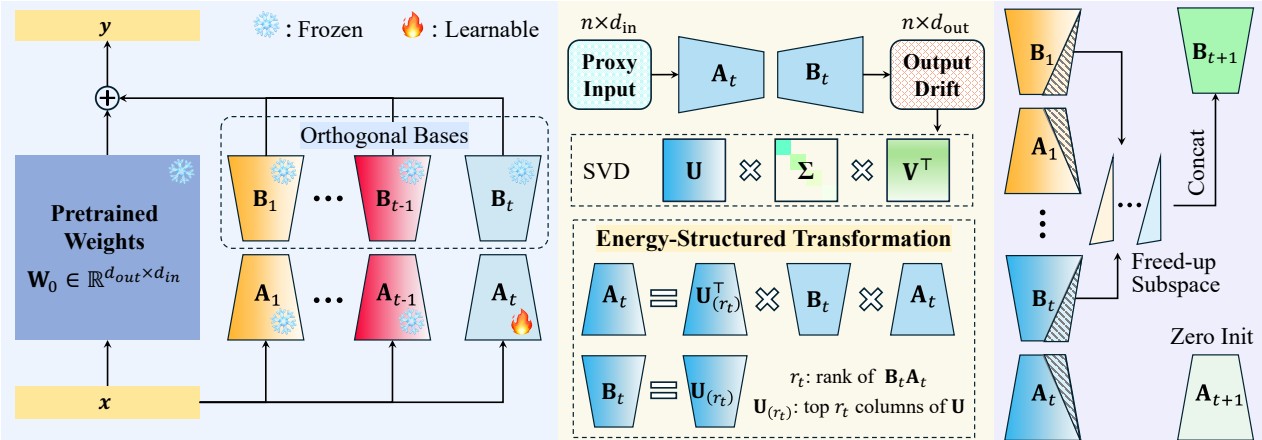

*Figure 2.* Overview of the proposed E$^2$-LoRA. Each new task is learned within a frozen orthogonal subspace to prevent interference with previous tasks. After task training, an energy-structured transformation reorganizes the LoRA parameters by rank-wise importance, enabling low-importance ranks to be discarded and reused for subsequent tasks.

where $r_t$ denotes the rank of the LoRA module $\mathbf{B}_t\mathbf{A}_t$. As a result, each rank component of LoRA becomes energy-ordered, and task-specific knowledge is concentrated in the leading ranks. Consequently, this construction enjoys a rank truncation optimality property, as formalized in Equation 3.1, and admits the expected output feature reconstruction error bound given in Equation 3.2.

**Proposition 3.1** (Optimality Property). *Among all parameter updates with rank at most $r$, $\mathbf{B}_t[:, : r]\mathbf{A}_t[: r, :]$ minimizes the expected output reconstruction error:*

$$\mathbb{E}_{\boldsymbol{x}\sim\mathcal{D}_t}\left\|\mathbf{B}_t[:, : r]\mathbf{A}_t[: r, :]\boldsymbol{x} - \mathbf{B}_t\mathbf{A}_t\boldsymbol{x}\right\|^2. \quad (7)$$

We provide a detailed proof in Appendix A.1.

**Proposition 3.2** (Truncation Error). *When truncating weight update matrix $\Delta\mathbf{W}_t = \mathbf{B}_t\mathbf{A}_t$ to retain only the leading $r$ ranks, i.e., $\Delta\mathbf{W}_t^{(r)} = \mathbf{B}_t[:, : r]\mathbf{A}_t[: r, :]$, the expected truncated output feature error is given by:*

$$\mathbb{E}_{\boldsymbol{x}\sim\mathcal{D}_t}\left[\|\Delta\mathbf{W}\boldsymbol{x} - \Delta\mathbf{W}_t^{(r)}\boldsymbol{x}\|_2^2\right] = \sum_{i=r+1}^{d_{out}}\sigma_i^2. \quad (8)$$

We provide a detailed proof in Appendix A.2.

This property implies that our energy-structured transformation maximally preserves task-specific knowledge while reducing rank, making it particularly suitable for continual learning.

**Output Feature Drift–Induced Orthogonalization.** The above transformation ensures that the knowledge of previous tasks is energy-ordered and concentrated in the leading ranks, allowing the remaining directions to be discarded

with negligible loss and reused for learning new tasks. Suppose that when task $t$ is about to be learned, the numbers of retained ranks for the previous tasks are $r_1^{(t)}, \cdots, r_{t-1}^{(t)}$ (the rank allocation strategy is introduced in a later section). We first prune the LoRA parameters of previous tasks as:

$$\mathbf{B}_k \leftarrow \mathbf{B}_k[:, : r_k^{(t)}], \quad \mathbf{A}_k \leftarrow \mathbf{A}_k[: r_k^{(t)}, :], \quad (9)$$

We then initialize the LoRA module for the new task as:

$$\mathbf{B}_t \leftarrow \left[\mathbf{B}_1[:, r_1^{(t)} :], \cdots, \mathbf{B}_2[:, r_2^{(t)} :]\right], \quad \mathbf{A}_t \leftarrow \mathbf{0}. \quad (10)$$

During the learning of task $t$, $\mathbf{B}_t$ is kept frozen, which naturally enforces orthogonality between the output directions of the new task and those of previous tasks, thereby preventing interference. After task $t$ is learned, the same output-drift-based transformation described in the previous sections is applied to $\mathbf{B}_t\mathbf{A}_t$, ensuring that the newly acquired knowledge is again energy-ordered and energy-concentrated. This iterative process progressively frees low-energy directions, providing sustainable capacity for subsequent tasks.

### 3.3. Dynamic Rank Allocation for Continual Learning

Building upon the proposed output feature drift–induced orthogonalization, we design a dynamic rank allocation strategy to manage the stability-plasticity trade-off within a bounded parameter space. We view the output dimension $d_{out}$ as a finite capacity pool shared across all learned tasks.

When a new task $t$ arrives, we first release capacity from previously learned tasks by pruning their low-energy LoRA ranks. Specifically, for each previous task $k < t$, let $\{\sigma_{k,i}^2\}_{i=1}^{r_k}$ denote the squared singular values (energy) of its output drift, ordered in descending order. We retain only

the smallest number of ranks $r_k^{(t)}$ such that the cumulative energy retention ratio satisfies:

$$\frac{\sum_{i=1}^{r_k^{(t)}} \sigma_{k,i}^2}{\sum_{i=1}^{r_k} \sigma_{k,i}^2} \geq \rho, \tag{11}$$

where $\rho \in (0,1)$ is a predefined energy retention threshold. The remaining low-energy ranks are discarded and their corresponding orthogonal subspaces are released for reuse by the new task. To ensure sufficient plasticity for learning task $t$, we enforce a minimum rank budget for the new task:

$$r_t^{\min} = \left\lceil \frac{d_{\text{out}}}{t} \right\rceil. \tag{12}$$

If the capacity freed by energy-based pruning is insufficient to meet this minimum requirement, we further prune the least important ranks from all existing tasks in a uniform manner—always removing the lowest-energy directions—until the available capacity reaches $r_t^{\min}$. Since ranks are ordered by energy and contribute minimally to the task output, this additional pruning incurs negligible degradation of previously learned tasks.

Overall, this dynamic rank allocation strategy enables adaptive capacity management in continual learning: important knowledge from previous tasks is preserved optimally, while low-energy directions are progressively reclaimed to provide sufficient learning space for new tasks.

### 3.4. Whole Process of E²-LoRA

The training pipeline is illustrated in Algorithm 1. When the $t$-th task arrives, E²-LoRA first allocates a task-specific learnable LoRA module following the dynamic rank allocation strategy described in Section 3.3. The backbone network and all LoRA components learned from previous tasks are kept frozen, while only the LoRA parameters associated with the current task are optimized. To further mitigate catastrophic forgetting, we additionally incorporate self-distillation and classifier alignment, two techniques commonly adopted in continual learning (Li & Hoiem, 2017; Zhang et al., 2023).

**Self-Distillation.** For tasks $t > 1$, we employ a self-distillation strategy to preserve knowledge of previously learned classes. Specifically, before updating the model on the current task, we treat the model with frozen backbone and historical LoRA parameters as a teacher, and obtain its predictions without gradient propagation. Let $z_{\text{tea}}$ denote the logits produced by the teacher model, and $z_{\text{stu}}$ denote the logits produced by the student model (i.e., the same network with the current task's LoRA activated). For the old classes, indexed by $\mathcal{C}_{\text{old}}$, we apply temperature-scaled

---

**Algorithm 1** E²-LoRA for Continual Learning

**Require:** Pre-trained weights $\mathbf{W}_0$, task datasets $\{\mathcal{D}_t\}_{t=1}^T$.
1: **for** task $t = 1$ to $T$ **do**
2:     **// Phase 1: Dynamic Rank Allocation**
3:     **if** $t > 1$ **then**
4:         **for** previous task $k = 1$ to $t - 1$ **do**
5:             Free-up orthogonal subspace (Eq. 9).
6:         **end for**
7:         Initialize $\mathbf{B}_t$ using the freed-up basis (Eq. 10).
8:     **else**
9:         Randomly initialize $\mathbf{B}_t$ with full rank.
10:    **end if**
11:    Initialize $\mathbf{A}_t$ to zero with the same rank as $\mathbf{B}_t$.
12:    **// Phase 2: Task Training**
13:    Optimize $\mathbf{A}_t$ on $\mathcal{D}_t$ minimizing $\mathcal{L}$ (Eq. 15).
14:    **// Phase 3: Energy-Structured Transformation**
15:    Sample proxy input feature batch $\mathbf{X}_t$.
16:    Compute output drift: $\Delta\mathbf{Y}_t = \mathbf{B}_t\mathbf{A}_t\mathbf{X}_t$ (Eq. 4).
17:    Perform SVD on drift: $\Delta\mathbf{Y}_t = \mathbf{U}_t\boldsymbol{\Sigma}_t\mathbf{V}_t^\top$ (Eq. 5).
18:    Transform $\mathbf{B}_t, \mathbf{A}_t$ using $\mathbf{U}_t$ (Eq. 6)
19:    **// Phase 4: Classifier Alignment**
20:    Fine-tune classifier heads using synthetic features generated from class statistics.
21: **end for**

---

distillation. The distillation loss is defined as:

$$\mathcal{L}_{\text{distill}} = T^2 \cdot \text{KL}\left( \text{Softmax}\left(\frac{z_{\mathcal{C}_{\text{old}}}^{\text{tea}}}{T}\right) \,\Big\|\, \text{Softmax}\left(\frac{z_{\mathcal{C}_{\text{old}}}^{\text{stu}}}{T}\right) \right), \tag{13}$$

where $T$ is the temperature parameter (set to $T = 2$ in our experiments), and $\text{KL}(\cdot\|\cdot)$ denotes the Kullback–Leibler divergence. For the new classes $\mathcal{C}_{\text{new}}$, the standard cross-entropy loss is applied:

$$\mathcal{L}_{\text{ce}} = \text{CE}(z_{\mathcal{C}_{\text{new}}}^{\text{stu}}, \boldsymbol{y}_{\mathcal{C}_{\text{new}}}), \tag{14}$$

where $\boldsymbol{y}_{\mathcal{C}_{\text{new}}}$ denotes the ground-truth labels of the current task. The overall training objective is:

$$\mathcal{L} = \mathcal{L}_{\text{ce}} + \lambda\mathcal{L}_{\text{distill}}, \tag{15}$$

where $\lambda$ controls the strength of distillation. This self-distillation mechanism encourages the student model to maintain consistent predictions on previously learned classes, thereby alleviating forgetting.

**Classifier Alignment.** After completing training on the $t$-th task, we further perform classifier alignment to reduce bias toward recently learned classes. Let $\mathcal{F}_t = \{\boldsymbol{r}_{i,t}\}_{i=1}^{n_t}$ denote the set of features extracted from the training samples of task $t$, where $\boldsymbol{r}_{i,t} = f_\Theta(\boldsymbol{x}_{i,t})$ and $f_\Theta(\cdot)$ represents the backbone network. For each class $c$ encountered during continual learning, we estimate the class-wise feature

*Table 1.* Average and last performance on class-incremental learning benchmarks with ViT-B/16-IN21K as the pre-trained backbone.

| Method | ImageNet-R | | CIFAR-100 | | CUB-200 | | Cars-196 | |
|---|---|---|---|---|---|---|---|---|
| | Last-Acc | Inc-Acc | Last-Acc | Inc-Acc | Last-Acc | Inc-Acc | Last-Acc | Inc-Acc |
| *Joint Training* | $82.76_{\pm 0.54}$ | - | $93.13_{\pm 0.21}$ | - | $88.26_{\pm 0.73}$ | - | $80.31_{\pm 0.13}$ | - |
| L2P (Wang et al., 2022b) | $66.49_{\pm 0.40}$ | $72.83_{\pm 0.56}$ | $82.76_{\pm 1.17}$ | $88.48_{\pm 0.83}$ | $62.21_{\pm 1.92}$ | $73.83_{\pm 1.67}$ | $38.18_{\pm 2.33}$ | $51.79_{\pm 4.19}$ |
| DualPrompt (Wang et al., 2022a) | $68.50_{\pm 0.52}$ | $72.59_{\pm 0.24}$ | $85.56_{\pm 0.33}$ | $90.33_{\pm 0.33}$ | $66.00_{\pm 0.57}$ | $77.92_{\pm 0.50}$ | $40.14_{\pm 2.36}$ | $56.74_{\pm 1.78}$ |
| SLCA (Zhang et al., 2023) | $77.00_{\pm 0.33}$ | $81.17_{\pm 0.64}$ | $91.53_{\pm 0.28}$ | $94.09_{\pm 0.87}$ | $84.71_{\pm 0.40}$ | $90.94_{\pm 0.68}$ | $67.73_{\pm 0.85}$ | $76.93_{\pm 1.21}$ |
| RanPAC (McDonnell et al., 2023) | $75.28_{\pm 0.14}$ | $80.66_{\pm 0.58}$ | $91.09_{\pm 0.25}$ | $94.03_{\pm 0.58}$ | $87.88_{\pm 0.53}$ | $92.57_{\pm 0.55}$ | $61.43_{\pm 0.84}$ | $72.36_{\pm 1.18}$ |
| SSIAT (Tan et al., 2024) | $79.38_{\pm 0.59}$ | $83.63_{\pm 0.43}$ | $91.35_{\pm 0.26}$ | $94.35_{\pm 0.60}$ | $88.75_{\pm 0.38}$ | $\mathbf{93.00}_{\pm 0.90}$ | $\underline{71.02}_{\pm 0.69}$ | $76.93_{\pm 0.62}$ |
| EASE (Zhou et al., 2024) | $77.75_{\pm 0.18}$ | $83.87_{\pm 0.78}$ | $86.54_{\pm 0.29}$ | $91.68_{\pm 0.18}$ | $79.90_{\pm 0.29}$ | $86.65_{\pm 0.63}$ | $35.46_{\pm 1.50}$ | $48.96_{\pm 0.96}$ |
| BiLoRA (Zhu et al., 2025) | $77.95_{\pm 0.14}$ | $81.52_{\pm 0.26}$ | $87.46_{\pm 0.76}$ | $92.50_{\pm 0.62}$ | - | - | - | - |
| MOS (Sun et al., 2025) | $78.10_{\pm 0.12}$ | $83.61_{\pm 0.40}$ | $90.53_{\pm 0.35}$ | $94.09_{\pm 0.91}$ | $\mathbf{89.91}_{\pm 0.25}$ | $\underline{92.87}_{\pm 0.56}$ | $67.76_{\pm 1.28}$ | $74.38_{\pm 1.13}$ |
| TUNA (Wang et al., 2025) | $\underline{79.44}_{\pm 0.38}$ | $\underline{84.80}_{\pm 0.37}$ | $\underline{91.79}_{\pm 0.15}$ | $\underline{94.88}_{\pm 0.06}$ | $88.40_{\pm 0.42}$ | $92.01_{\pm 0.68}$ | $69.46_{\pm 0.35}$ | $\underline{76.95}_{\pm 0.39}$ |
| E$^2$-LoRA (Ours) | $\mathbf{82.77}_{\pm 0.10}$ | $\mathbf{87.18}_{\pm 0.12}$ | $\mathbf{92.13}_{\pm 0.19}$ | $\mathbf{95.01}_{\pm 0.02}$ | $\underline{89.77}_{\pm 0.16}$ | $92.68_{\pm 0.48}$ | $\mathbf{75.82}_{\pm 0.28}$ | $\mathbf{80.90}_{\pm 0.73}$ |

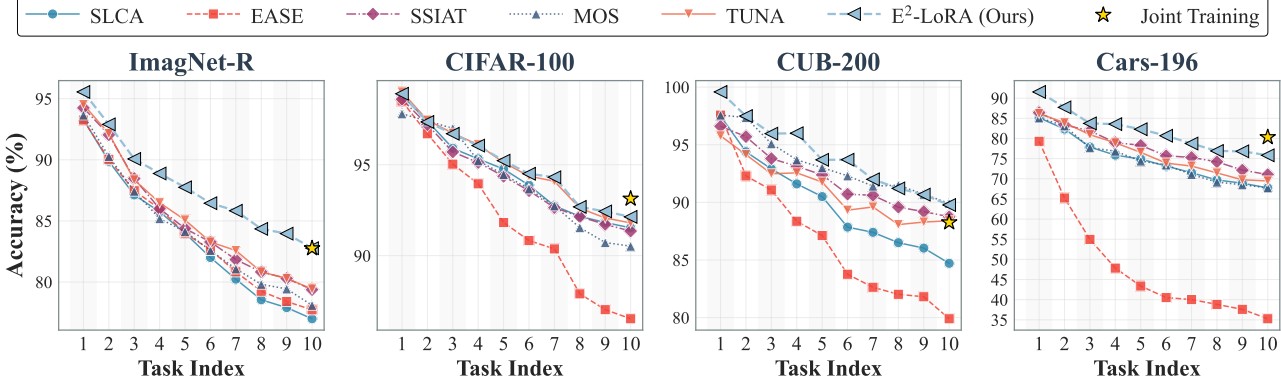

*Figure 3.* Evolution of accuracy across sequential tasks with ViT-B/16-IN21K as the pre-trained backbone.

distribution by computing the empirical mean $\boldsymbol{\mu}_c$ and co-variance $\boldsymbol{\Sigma}_c$. We then sample $S$ synthetic features from the Gaussian distribution $\boldsymbol{r} \sim \mathcal{N}(\boldsymbol{\mu}_c, \boldsymbol{\Sigma}_c)$. Finally, we fine-tune the classifier using these generated samples and the standard cross-entropy loss. This alignment step rebalances the classifier across old and new classes, effectively mitigating classifier bias induced by task-wise incremental training.

## 4. Experiments

### 4.1. Experimental Setup

**Datasets.** We conduct comprehensive experiments on four widely-used class-incremental learning benchmarks: ImageNet-R (Hendrycks et al., 2021), CIFAR-100 (Krizhevsky et al., 2009), CUB-200 (Wah et al., 2011), and Cars-196 (Krause et al., 2013). Following (Zhang et al., 2023), we split each dataset into 10 sequential tasks. We also conduct experiments on two domain-incremental learning benchmarks, Office-Home (Venkateswara et al., 2017) and DomainNet (Peng et al., 2019).

**Evaluation Metrics.** We adopt two standard evaluation protocols to measure both stability and plasticity. The Last-Accuracy (Last-Acc) metric represents the final average accuracy on all seen tasks after learning the last task, measuring overall performance. The Incremental Accuracy (Inc-Acc) metric indicates the average accuracy across all incremental steps, calculated as $\frac{1}{T}\sum_{t=1}^{T} A_t$ where $T$ is the total number of tasks and $A_t$ is the accuracy after learning task $t$.

**Implementation Details.** We consider two typical pre-trained weights, namely ViT-B/16-IN21K and ViT-B/16-IN1K. Both are initially pre-trained on ImageNet-21K (Ridnik et al., 2021), with the latter further fine-tuned on ImageNet-1K. The learning rate for the classification layer is set to 0.01, while that for LoRA is configured at 0.0005. We employ the SGD optimizer with a batch size of 64, and the weight for the distillation loss is set to $\lambda = 0.2$. The energy retention ratio threshold is set to $\rho = 0.9999$. We conduct three independent runs for each experiment and report the mean and standard deviation of the results.

*Table 2.* Average and last performance comparison on longer-term benchmarks with ViT-B/16-IN21K as the pre-trained backbone.

| Method | ImageNet-R 20 tasks | | ImageNet-R 50 tasks | | CIFAR-100 20 tasks | | CIFAR-100 50 tasks | |
|---|---|---|---|---|---|---|---|---|
| | Last-Acc | Inc-Acc | Last-Acc | Inc-Acc | Last-Acc | Inc-Acc | Last-Acc | Inc-Acc |
| *Joint Training* | $82.76_{\pm0.54}$ | - | $82.76_{\pm0.54}$ | - | $93.13_{\pm0.21}$ | - | $93.13_{\pm0.21}$ | - |
| L2P (Wang et al., 2022b) | $62.15_{\pm1.17}$ | $68.35_{\pm2.12}$ | $55.89_{\pm1.59}$ | $62.98_{\pm2.89}$ | $80.72_{\pm1.12}$ | $87.18_{\pm0.83}$ | $73.91_{\pm1.67}$ | $81.90_{\pm0.98}$ |
| DualPrompt (Wang et al., 2022a) | $66.89_{\pm0.40}$ | $73.07_{\pm1.21}$ | $61.50_{\pm0.86}$ | $68.63_{\pm1.31}$ | $83.82_{\pm0.51}$ | $90.22_{\pm0.68}$ | $76.66_{\pm0.74}$ | $85.18_{\pm0.92}$ |
| CODA-Prompt (Smith et al., 2023) | $67.53_{\pm0.24}$ | $73.64_{\pm0.95}$ | $48.89_{\pm0.90}$ | $55.59_{\pm2.67}$ | $81.19_{\pm0.31}$ | $87.27_{\pm0.35}$ | $55.45_{\pm0.48}$ | $68.39_{\pm0.53}$ |
| InfLoRA (Liang & Li, 2024) | $71.46_{\pm0.95}$ | $78.32_{\pm1.22}$ | $60.49_{\pm1.43}$ | $69.95_{\pm2.18}$ | $82.19_{\pm1.33}$ | $88.05_{\pm0.64}$ | $56.65_{\pm5.55}$ | $70.29_{\pm1.86}$ |
| EASE (Zhou et al., 2024) | $73.78_{\pm0.44}$ | $80.29_{\pm0.72}$ | $68.54_{\pm0.71}$ | $75.77_{\pm0.58}$ | $82.21_{\pm0.72}$ | $90.76_{\pm0.54}$ | $82.10_{\pm0.66}$ | $87.65_{\pm0.46}$ |
| LORA$^-$DRS (Liu & Chang, 2025) | $74.80_{\pm0.73}$ | $80.69_{\pm0.75}$ | $72.12_{\pm0.87}$ | $77.94_{\pm0.74}$ | $88.69_{\pm0.15}$ | $92.25_{\pm0.24}$ | $87.29_{\pm0.31}$ | $91.29_{\pm0.29}$ |
| MOS (Sun et al., 2025) | $75.58_{\pm0.47}$ | $81.76_{\pm0.26}$ | $77.29_{\pm0.77}$ | $83.06_{\pm1.41}$ | $88.85_{\pm0.43}$ | $92.86_{\pm0.37}$ | $85.51_{\pm0.04}$ | $91.00_{\pm0.26}$ |
| TUNA (Wang et al., 2025) | $77.32_{\pm0.31}$ | $83.05_{\pm0.64}$ | $75.35_{\pm0.67}$ | $80.95_{\pm1.29}$ | $91.31_{\pm0.12}$ | $94.68_{\pm0.16}$ | $89.41_{\pm0.23}$ | $93.00_{\pm0.42}$ |
| E$^2$-LoRA (Ours) | $80.77_{\pm0.13}$ | $86.37_{\pm0.39}$ | $78.58_{\pm0.22}$ | $83.96_{\pm0.76}$ | $91.42_{\pm0.10}$ | $94.55_{\pm0.07}$ | $90.70_{\pm0.29}$ | $93.86_{\pm0.09}$ |

*Table 3.* Average and last performance on domain-incremental learning benchmarks with ViT-B/16-IN1K as the backbone.

| DS | Method | Last-Acc | Inc-Acc |
|---|---|---|---|
| Office-Home | L2P (Wang et al., 2022b) | $80.03_{\pm1.29}$ | $79.72_{\pm4.19}$ |
| | DualPrompt (Wang et al., 2022a) | $80.85_{\pm0.14}$ | $80.20_{\pm3.81}$ |
| | CODA-Prompt (Smith et al., 2023) | $85.07_{\pm0.34}$ | $84.70_{\pm2.94}$ |
| | MEMO (Zhou et al., 2023) | $63.09_{\pm1.80}$ | $71.18_{\pm2.76}$ |
| | RanPAC (McDonnell et al., 2023) | $82.28_{\pm0.07}$ | $82.30_{\pm3.34}$ |
| | EASE (Zhou et al., 2024) | $76.33_{\pm2.16}$ | $81.16_{\pm3.52}$ |
| | SimpleCIL (Zhou et al., 2025a) | $75.72_{\pm0.00}$ | $75.69_{\pm5.03}$ |
| | DCE (Li et al., 2025b) | $84.40_{\pm0.20}$ | $84.60_{\pm3.00}$ |
| | DUCT (Zhou et al., 2025b) | $85.42_{\pm0.33}$ | $81.28_{\pm0.31}$ |
| | E$^2$-LoRA (Ours) | $88.25_{\pm0.25}$ | $84.94_{\pm0.21}$ |
| DomainNet | L2P (Wang et al., 2022b) | $48.72_{\pm2.83}$ | $50.45_{\pm4.10}$ |
| | DualPrompt (Wang et al., 2022a) | $50.46_{\pm3.17}$ | $52.28_{\pm3.35}$ |
| | CODA-Prompt (Smith et al., 2023) | $59.99_{\pm0.88}$ | $59.85_{\pm4.49}$ |
| | MEMO (Zhou et al., 2023) | $58.41_{\pm3.20}$ | $61.92_{\pm5.39}$ |
| | RanPAC (McDonnell et al., 2023) | $54.80_{\pm0.36}$ | $55.20_{\pm3.93}$ |
| | EASE (Zhou et al., 2024) | $43.72_{\pm1.70}$ | $50.50_{\pm2.27}$ |
| | SimpleCIL (Zhou et al., 2025a) | $44.08_{\pm0.00}$ | $42.95_{\pm4.84}$ |
| | DCE (Li et al., 2025b) | $63.50_{\pm0.50}$ | $64.30_{\pm6.00}$ |
| | DUCT (Zhou et al., 2025b) | $67.01_{\pm1.35}$ | $67.16_{\pm3.75}$ |
| | E$^2$-LoRA (Ours) | $69.63_{\pm0.05}$ | $68.69_{\pm0.04}$ |

## 4.2. Main Results

**Class-Incremental Learning Results.** Table 1 presents a comprehensive performance comparison on four widely-used benchmarks under a 10-task incremental setting. Figure 3 illustrates the evolution of accuracy across sequential tasks. Across all datasets, E$^2$-LoRA consistently outperforms existing state-of-the-art (SOTA) methods. Most notably, our method achieves results that are remarkably close to—and in some instances, even surpass—the Joint Training upper bound. For example, on ImageNet-R, E$^2$-LoRA attains a Last-Acc of 82.77%, effectively matching the Joint Training performance of 82.76%. On Cars-196, we achieve 75.82% Last-Acc, drastically reducing the gap to the up-

per bound compared to prior arts like TUNA (69.46%). This phenomenon suggests that our energy-concentrated subspace representation not only prevents catastrophic forgetting but also facilitates a "denoising" effect that focuses the model on the most transferable spectral features, occasionally yielding better generalization than the standard full-rank joint optimization.

**Domain-Incremental Learning Results.** Table 3 reports the performance of domain-incremental learning on two widely used benchmarks, Office-Home and DomainNet. E$^2$-LoRA consistently outperforms existing state-of-the-art methods by a clear margin on both benchmarks. These results demonstrate that our approach effectively isolates domain-specific information in orthogonal, energy-ordered subspaces, thereby preventing interference across domains and enabling robust continual adaptation.

**Robustness in Long-Term Continual Learning.** To further evaluate the scalability of our approach, Table 2 summarizes performance under more demanding, longer-term scenarios involving 20 and 50 sequential tasks. As the number of tasks increases, most baseline methods suffer from a severe performance collapse due to the accumulation of task interference and capacity saturation. In contrast, E$^2$-LoRA maintains highly stable and superior accuracy even in the 50-task setting. Specifically, on ImageNet-R 50-tasks, E$^2$-LoRA preserves a Last-Acc of 78.58%, outperforming the competitive TUNA baseline by over 3%. Similarly, on CIFAR-100 50-tasks, we maintain a robust 90.70% Last-Acc. These results demonstrate that our dynamic, energy-ordered rank allocation effectively mitigates the stability-plasticity dilemma; by prioritizing high-energy components and gracefully pruning less significant ones, E$^2$-LoRA sustains a near-constant learning capacity over extended horizons.

*Table 4.* Ablation study on the contribution of individual components in the proposed method.

| Method | KD | CA | ImageNet-R | | CIFAR-100 | | CUB-200 | | Cars-196 | |
|---|---|---|---|---|---|---|---|---|---|---|
| | | | Last-Acc | Inc-Acc | Last-Acc | Inc-Acc | Last-Acc | Inc-Acc | Last-Acc | Inc-Acc |
| *Joint Training* | - | - | $82.76_{\pm0.54}$ | - | $93.13_{\pm0.21}$ | - | $88.26_{\pm0.73}$ | - | $80.31_{\pm0.13}$ | - |
| O-LoRA | ✗ | ✗ | $75.38_{\pm0.39}$ | $80.53_{\pm0.24}$ | $88.34_{\pm0.16}$ | $92.45_{\pm0.16}$ | $82.49_{\pm0.58}$ | $89.31_{\pm0.37}$ | $54.26_{\pm0.88}$ | $62.94_{\pm3.54}$ |
| | ✓ | ✗ | $75.37_{\pm0.51}$ | $80.42_{\pm0.26}$ | $88.16_{\pm0.50}$ | $92.21_{\pm0.16}$ | $82.40_{\pm0.26}$ | $89.27_{\pm0.06}$ | $52.47_{\pm1.00}$ | $61.62_{\pm1.04}$ |
| | ✗ | ✓ | $77.88_{\pm0.41}$ | $82.39_{\pm0.27}$ | $90.78_{\pm0.28}$ | $93.62_{\pm0.27}$ | $88.89_{\pm0.75}$ | $92.52_{\pm0.35}$ | $65.67_{\pm0.42}$ | $72.73_{\pm0.49}$ |
| | ✓ | ✓ | $77.92_{\pm0.61}$ | $82.25_{\pm0.40}$ | $90.72_{\pm0.12}$ | $93.61_{\pm0.20}$ | $89.18_{\pm0.38}$ | $92.91_{\pm0.68}$ | $67.61_{\pm0.85}$ | $73.51_{\pm0.40}$ |
| InfLoRA | ✗ | ✗ | $77.59_{\pm0.52}$ | $83.45_{\pm0.31}$ | $85.20_{\pm0.31}$ | $90.57_{\pm0.25}$ | $79.89_{\pm0.45}$ | $88.26_{\pm0.12}$ | $57.66_{\pm0.65}$ | $70.03_{\pm0.45}$ |
| | ✓ | ✗ | $77.44_{\pm0.37}$ | $84.10_{\pm0.22}$ | $85.57_{\pm0.09}$ | $91.15_{\pm0.18}$ | $80.99_{\pm0.60}$ | $88.90_{\pm0.41}$ | $63.32_{\pm0.92}$ | $72.63_{\pm0.80}$ |
| | ✗ | ✓ | $78.00_{\pm0.49}$ | $84.43_{\pm0.28}$ | $86.96_{\pm0.22}$ | $92.01_{\pm0.29}$ | $81.35_{\pm0.33}$ | $88.93_{\pm0.35}$ | $58.95_{\pm0.54}$ | $72.08_{\pm0.20}$ |
| | ✓ | ✓ | $77.87_{\pm0.61}$ | $85.09_{\pm0.35}$ | $87.33_{\pm0.45}$ | $92.60_{\pm0.21}$ | $82.18_{\pm0.71}$ | $89.91_{\pm0.58}$ | $69.73_{\pm0.45}$ | $78.60_{\pm0.17}$ |
| $E^2$-LoRA (Ours) | ✗ | ✗ | $80.28_{\pm0.15}$ | $85.65_{\pm0.39}$ | $89.42_{\pm0.31}$ | $93.47_{\pm0.21}$ | $81.80_{\pm0.33}$ | $89.14_{\pm0.49}$ | $66.04_{\pm0.26}$ | $75.18_{\pm0.27}$ |
| | ✓ | ✗ | $80.28_{\pm0.24}$ | $85.47_{\pm0.29}$ | $89.03_{\pm0.46}$ | $93.03_{\pm0.36}$ | $83.04_{\pm0.19}$ | $89.67_{\pm0.26}$ | $63.34_{\pm0.44}$ | $72.64_{\pm0.16}$ |
| | ✗ | ✓ | $82.52_{\pm0.31}$ | $87.11_{\pm0.45}$ | $92.02_{\pm0.08}$ | $94.88_{\pm0.08}$ | $89.29_{\pm0.25}$ | $92.33_{\pm0.41}$ | $72.15_{\pm0.20}$ | $78.49_{\pm0.49}$ |
| | ✓ | ✓ | $\mathbf{82.77}_{\pm0.10}$ | $\mathbf{87.18}_{\pm0.12}$ | $\mathbf{92.13}_{\pm0.19}$ | $\mathbf{95.01}_{\pm0.02}$ | $\mathbf{89.77}_{\pm0.16}$ | $\mathbf{92.68}_{\pm0.48}$ | $\mathbf{75.82}_{\pm0.28}$ | $\mathbf{80.90}_{\pm0.73}$ |

*Table 5.* Effect analysis of the proposed rank allocation strategy.

| Method | ImageNet-R 10 tasks | | ImageNet-R 50 tasks | |
|---|---|---|---|---|
| | Last-Acc | Inc-Acc | Last-Acc | Inc-Acc |
| *Full Rank* | $80.98_{\pm0.34}$ | $86.48_{\pm0.51}$ | $68.52_{\pm0.73}$ | $80.25_{\pm1.13}$ |
| $d_{out}/5$ | $82.01_{\pm0.32}$ | $86.08_{\pm0.72}$ | $77.90_{\pm0.14}$ | $82.82_{\pm0.96}$ |
| $d_{out}/10$ | $81.20_{\pm0.09}$ | $84.84_{\pm0.74}$ | $76.69_{\pm0.83}$ | $81.57_{\pm1.13}$ |
| $d_{out}/50$ | $77.71_{\pm0.10}$ | $81.19_{\pm0.93}$ | $73.65_{\pm0.10}$ | $78.52_{\pm1.14}$ |
| Ours | $\mathbf{82.77}_{\pm0.10}$ | $\mathbf{87.18}_{\pm0.12}$ | $\mathbf{78.58}_{\pm0.22}$ | $\mathbf{83.96}_{\pm0.76}$ |

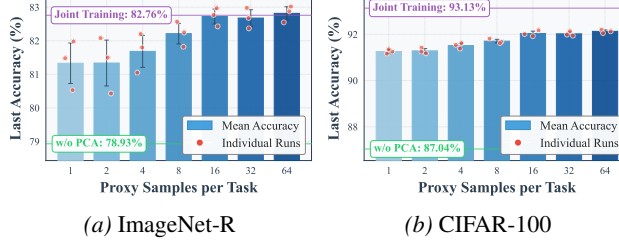

*(a) ImageNet-R*  *(b) CIFAR-100*

*Figure 4.* Impact of the number of proxy samples used in $E^2$-LoRA. Each configuration is evaluated over three random runs.

### 4.3. Ablation and Analysis

**Ablation Study.** We conduct comprehensive ablation and replacement experiments on each component of our method, as summarized in Table 4. Specifically, we replace our proposed $E^2$-LoRA with alternative backbone adaptation methods, including O-LoRA and InfLoRA, and evaluate their performance under the same continual learning setting. The results show that $E^2$-LoRA achieves significantly superior performance, demonstrating that energy ordering and concentration of LoRA ranks effectively balance the stability of previously learned knowledge and the plasticity required for learning new tasks. In addition, incorporating simple yet effective knowledge distillation (KD) and classifier alignment (CA) further improves performance, enabling our method to approach the upper bound achieved by joint training.

**Effect of the Rank Allocation Strategy.** Table 5 illustrates the impact of allocating different ranks to new tasks on continual learning performance. The results show that fixed rank allocation strategies struggle to achieve optimal performance. In contrast, our method dynamically allocates ranks by jointly considering the energy retention of previous tasks and the plasticity required for the new task, consistently yielding superior results across settings.

**Effect of the Proxy Dataset Size.** We analyze the number of samples required in the proxy dataset for Output Feature Drift–Induced Orthogonalization. Figure 4 shows the effect of different proxy dataset sizes on the final performance. The results indicate that as few as 16 samples are sufficient to obtain a stable, energy-ordered and energy-concentrated orthogonal basis. This entails a very lightweight computational overhead, which can be performed immediately after task learning and does not require storing the proxy samples.

### 5. Conclusion

In this paper, we propose $E^2$-LoRA, a novel framework for continual learning that addresses the stability-plasticity dilemma by leveraging the spectral properties of task-specific output feature drift. Unlike prior methods that orthogonalize in parameter or input spaces, we demonstrate that task-specific knowledge is highly concentrated within a

low-dimensional subspace of the output drift. By applying an energy-structured transformation, $E^2$-LoRA explicitly orders knowledge by rank, enabling optimal preservation of past tasks while repurposing low-energy capacity for subsequent learning. Coupled with a dynamic rank allocation strategy, $E^2$-LoRA enables adaptive capacity management within a bounded parameter budget. Extensive evaluations across multiple benchmarks demonstrate that $E^2$-LoRA consistently achieves state-of-the-art performance, significantly mitigating catastrophic forgetting while maintaining high plasticity for future task adaptation.

## Impact Statement

This paper presents work whose goal is to advance the field of Machine Learning. There are many potential societal consequences of our work, none which we feel must be specifically highlighted here.

## Acknowledgement

This research was supported by the Jiangsu Science Foundation (BG2024036, BK20243012), the National Science Foundation of China (62125602, U24A20324, 92464301), the New Cornerstone Science Foundation through the XPLORER PRIZE, the Fundamental Research Funds for the Central Universities (2242025K30024), and the Big Data Computing Center of Southeast University.

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

# A. Proofs

This section provides the proofs for the optimality property under rank truncation and the expected output error after truncation.

## A.1. Optimality Property under Rank Truncation

**Proposition A.1** (Optimality Property). *Among all parameter updates with rank at most $r$, $\mathbf{B}_t[:,:r]\mathbf{A}_t[:r,:]$ minimizes the expected output reconstruction error:*

$$\mathbb{E}_{\boldsymbol{x}\sim\mathcal{D}_t}\left\|\mathbf{B}_t[:,:r]\mathbf{A}_t[:r,:]\boldsymbol{x} - \mathbf{B}_t\mathbf{A}_t\boldsymbol{x}\right\|^2. \quad (16)$$

*Proof.* For notational convenience, we denote $\Delta\mathbf{W}_t = \mathbf{B}_t\mathbf{A}_t$ and $\Delta\mathbf{W}_t^{(r)} = \mathbf{B}_t[:,:r]\mathbf{A}_t[:r,:]$. For any rank-$r$ parameter update $\widetilde{\Delta\mathbf{W}}$, the expected output reconstruction error can be written as:

$$\mathbb{E}_{\boldsymbol{x}\sim\mathcal{D}_t}\left\|\Delta\mathbf{W}_t\boldsymbol{x} - \widetilde{\Delta\mathbf{W}}\boldsymbol{x}\right\|^2$$
$$= \mathbb{E}_{\boldsymbol{x}\sim\mathcal{D}_t}\left\|(\Delta\mathbf{W}_t - \widetilde{\Delta\mathbf{W}})\boldsymbol{x}\right\|^2. \quad (17)$$

Let $\boldsymbol{\Sigma}_x = \mathbb{E}[\boldsymbol{x}\boldsymbol{x}^\top]$ denote the input covariance matrix. Then the expectation in (17) can be expressed as:

$$\mathbb{E}_{\boldsymbol{x}\sim\mathcal{D}_t}\left\|(\Delta\mathbf{W}_t - \widetilde{\Delta\mathbf{W}})\boldsymbol{x}\right\|^2$$
$$= \left\|(\Delta\mathbf{W}_t - \widetilde{\Delta\mathbf{W}})\boldsymbol{\Sigma}_x^{1/2}\right\|_F^2. \quad (18)$$

In practice, $\boldsymbol{\Sigma}_x$ is estimated using a proxy set $\mathbf{X}_t$, yielding:

$$\Delta\mathbf{Y}_t = \Delta\mathbf{W}_t\mathbf{X}_t. \quad (19)$$

Thus, minimizing the expected reconstruction error over all rank-$r$ updates $\widetilde{\Delta\mathbf{W}}$ is equivalent to solving:

$$\min_{\text{rank}(\widetilde{\Delta\mathbf{W}})\leq r}\left\|\Delta\mathbf{Y}_t - \widetilde{\Delta\mathbf{W}}\mathbf{X}_t\right\|_F^2. \quad (20)$$

Let the singular value decomposition of $\Delta\mathbf{Y}_t$ be:

$$\Delta\mathbf{Y}_t = \mathbf{U}_t\boldsymbol{\Sigma}_t\mathbf{V}_t^\top. \quad (21)$$

By the Eckart–Young–Mirsky theorem, the optimal rank-$r$ approximation that minimizes (20) is given by:

$$\Delta\mathbf{Y}_t^{(r)} = \mathbf{U}_t^{(r)}\boldsymbol{\Sigma}_t^{(r)}(\mathbf{V}_t^{(r)})^\top, \quad (22)$$

where $\mathbf{U}_t^{(r)}$ contains the top-$r$ left singular vectors. Accordingly, the corresponding parameter update:

$$\Delta\mathbf{W}_t^{(r)} = \mathbf{U}_t^{(r)}(\mathbf{U}_t^{(r)})^\top\Delta\mathbf{W}_t, \quad (23)$$

induces exactly $\Delta\mathbf{Y}_t^{(r)}$ and achieves the minimum output feature reconstruction error among all rank-$r$ updates. Therefore, $\Delta\mathbf{W}_t^{(r)}$ minimizes:

$$\mathbb{E}_{\boldsymbol{x}\sim\mathcal{D}_t}\left\|\Delta\mathbf{W}_t\boldsymbol{x} - \Delta\mathbf{W}_t^{(r)}\boldsymbol{x}\right\|^2, \quad (24)$$

over all rank-$r$ parameter updates, completing the proof. $\square$

## A.2. Output Truncation Error

**Proposition A.2** (Truncation Error). *When truncating weight update matrix $\Delta\mathbf{W}_t = \mathbf{B}_t\mathbf{A}_t$ to retain only the leading $r$ ranks, i.e., $\Delta\mathbf{W}_t^{(r)} = \mathbf{B}_t[:,:r]\mathbf{A}_t[:r,:]$, the expected truncated output feature error is given by:*

$$\mathbb{E}_{\boldsymbol{x}\sim\mathcal{D}_t}\left[\|\Delta\mathbf{W}\boldsymbol{x} - \Delta\mathbf{W}_t^{(r)}\boldsymbol{x}\|_2^2\right] = \sum_{i=r+1}^{d_{out}}\sigma_i^2. \quad (25)$$

*Proof.* For simplicity of exposition, we omit the task index $t$ in the following proof. The expected $L_2$ error over the input distribution $\mathcal{D}$ can be empirically estimated using the proxy set $\mathbf{X}$ (where $N$ is the number of samples):

$$\mathcal{L} = \mathbb{E}_{x\sim\mathcal{D}}[\|(\Delta\mathbf{W} - \Delta\mathbf{W}^{(r)})x\|_2^2]$$
$$\approx \frac{1}{N}\sum_{n=1}^{N}\|(\Delta\mathbf{W} - \Delta\mathbf{W}^{(r)})x_n\|_2^2 \quad (26)$$
$$= \frac{1}{N}\|(\Delta\mathbf{W} - \Delta\mathbf{W}^{(r)})\mathbf{X}\|_F^2.$$

Recall that the actual output drift is $\Delta\mathbf{Y} = \Delta\mathbf{W}\mathbf{X}$. The output of the rank-$r$ approximated model is $\Delta\mathbf{Y}^{(r)} = \Delta\mathbf{W}^{(r)}\mathbf{X}$. Thus:

$$\mathcal{L} = \frac{1}{N}\|\Delta\mathbf{Y} - \Delta\mathbf{Y}^{(r)}\|_F^2. \quad (27)$$

In $\text{E}^2$-LoRA, the approximation is constructed by projecting the original drift onto the subspace spanned by the first $r$ principal components of the output space:

$$\Delta\mathbf{Y}^{(r)} = \mathbf{U}^{(r)}(\mathbf{U}^{(r)})^\top\Delta\mathbf{Y}. \quad (28)$$

Substituting the full SVD $\Delta\mathbf{Y} = \mathbf{U}\boldsymbol{\Sigma}\mathbf{V}^\top$ into this expression:

$$\Delta\mathbf{Y}^{(r)} = \mathbf{U}^{(r)}(\mathbf{U}^{(r)})^\top(\mathbf{U}\boldsymbol{\Sigma}\mathbf{V}^\top). \quad (29)$$

Since $\mathbf{U}$ is orthonormal ($\mathbf{U}^\top\mathbf{U} = \mathbf{I}$), the term $(\mathbf{U}^{(r)})^\top\mathbf{U}$ yields a truncated identity matrix $[\mathbf{I}_{r\times r} \mid \mathbf{0}_{r\times(d_{out}-r)}]$. Therefore:

$$\Delta\mathbf{Y}^{(r)} = \mathbf{U}\boldsymbol{\Sigma}_r\mathbf{V}^\top, \quad (30)$$

where $\boldsymbol{\Sigma}_r = \text{diag}(\sigma_1,\ldots,\sigma_r,0,\ldots,0)$. The error matrix (residual) is:

$$\mathbf{E}_r = \Delta\mathbf{Y} - \Delta\mathbf{Y}^{(r)} = \mathbf{U}(\boldsymbol{\Sigma} - \boldsymbol{\Sigma}_r)\mathbf{V}^\top = \sum_{i=r+1}^{d_{out}}\sigma_i u_i v_i^\top. \quad (31)$$

Using the fact that $\|\mathbf{U}\mathbf{A}\mathbf{V}^\top\|_F^2 = \|\mathbf{A}\|_F^2$ for orthonormal matrices $\mathbf{U}, \mathbf{V}$:

$$\|\mathbf{E}_r\|_F^2 = \|\mathbf{\Sigma} - \mathbf{\Sigma}_r\|_F^2 = \sum_{i=r+1}^{d_{\text{out}}} \sigma_i^2. \tag{32}$$

Thus:

$$\mathcal{L} = \frac{1}{N} \sum_{i=r+1}^{d_{\text{out}}} \sigma_i^2. \tag{33}$$

If we consider the singular values normalized by the sample size, the factor $1/N$ is absorbed, yielding:

$$\mathbb{E}_{x \sim \mathcal{D}} \left[ \|\Delta\mathbf{W}x - \Delta\mathbf{W}^{(r)}x\|_2^2 \right] = \sum_{i=r+1}^{d_{\text{out}}} \sigma_i^2. \tag{34}$$

This completes the proof.

$\square$

## B. Experimental Details

### B.1. Detailed Experimental Settings

Table 6 summarizes the experimental settings across different continual learning benchmarks, including the pre-trained models used, learning rates for the classifier and LoRA modules, the number of training epochs per task, the number of epochs for classifier alignment, and the batch size.

### B.2. Placement of LoRA Modules

In all experiments with $E^2$-LoRA, we integrate LoRA modules into the QKV projection layer of the Attention and the first layer of the MLP. This configuration is informed by our post-training energy analysis of model parameters, where we observed that the energy of parameter updates in the Attention O-projection and the second MLP layer is negligible. By omitting LoRA in these layers, we significantly reduce the computational overhead during training while maintaining competitive performance. During inference, the learned LoRA weights can be seamlessly merged into the original pre-trained parameters; thus, our method introduces zero additional inference latency or computational cost.

### B.3. Evaluation Strategy on Domain-Incremental Learning Benchmarks

In Domain-Incremental Learning (DIL), each task consists of samples from different domains belonging to a shared set of classes. To evaluate performance, we aggregate the prediction scores for each class by summing the outputs from the corresponding classifiers across all domains. The resulting cumulative score is then used to determine the final classification result for each category.

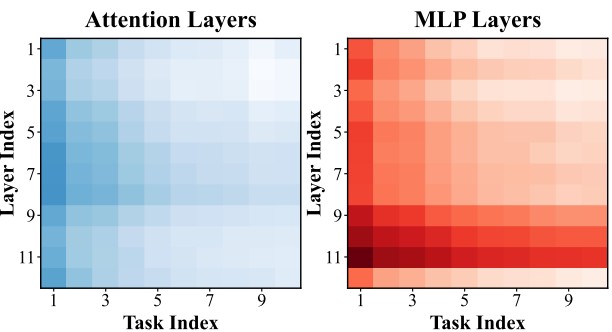

*Figure 5.* Changes in the energy of parameter updates as new tasks arrive. Experiments are conducted on the ImageNet-R dataset.

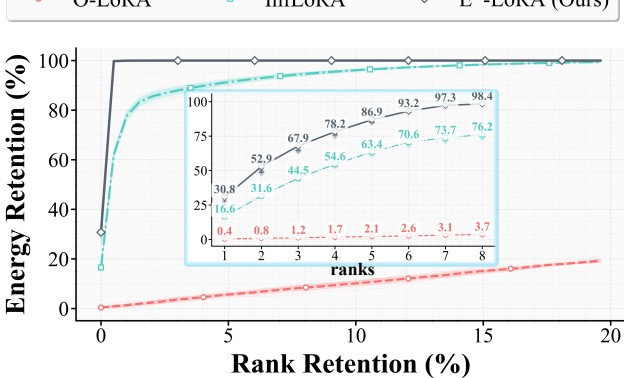

*Figure 6.* Energy Retention (%) vs. Rank Retention (%). Comparison of energy concentration across three representative methods: parameter bases (O-LoRA), principal bases of the input space (InfLoRA), and principal bases of the output drift space ($E^2$-LoRA).

## C. Additional Experiments

### C.1. Energy Evolution across Sequential Tasks.

We analyze the energy distribution of sequential tasks in Figure 5. Specifically, we conduct experiments on the ImageNet-R benchmark with 10 sequential tasks and record the singular values associated with the retained ranks of the LoRA modules for each task. The energy of a task is computed as the sum of the singular values over its allocated ranks. As the model learns classes in a sequential manner, the results reveal a clear trend: earlier tasks exhibit higher energy, while the energy gradually decreases as training progresses. This behavior indicates that the model progressively adapts to the shared data distribution across tasks, leading to more compact and efficient representations. Eventually, the energy converges to a low-loss region that reflects an optimized representation for the entire benchmark.

### C.2. Analysis of Energy Concentration

We analyze the energy concentration of different orthogonalization targets: the columns of weight matrices in O-LoRA,

*Table 6.* Continual learning settings across different benchmarks. CA denotes classifier alignment.

| Benchmarks | backbone | lr (classifier) | lr (LoRA) | epochs/task | CA epochs/task | batchsize |
|---|---|---|---|---|---|---|
| ImageNet-R (Hendrycks et al., 2021) | ViT-B/16-IN21K | 0.01 | 0.0005 | 5 | 3 | 64 |
| CIFAR-100 (Krizhevsky et al., 2009) | ViT-B/16-IN21K | 0.01 | 0.0005 | 5 | 3 | 64 |
| CUB-200 (Wah et al., 2011) | ViT-B/16-IN21K | 0.01 | 0.0005 | 10 | 3 | 64 |
| Cars-196 (Krause et al., 2013) | ViT-B/16-IN21K | 0.01 | 0.0005 | 50 | 3 | 64 |
| Office-Home (Venkateswara et al., 2017) | ViT-B/16-IN1K | 0.1 | 0.0005 | 15 | 5 | 64 |
| DomainNet (Peng et al., 2019) | ViT-B/16-IN1K | 0.1 | 0.0005 | 15 | 5 | 64 |

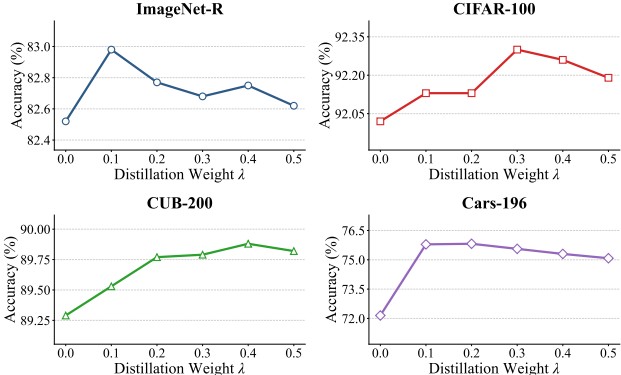

*Figure 7.* Impact of the distillation loss weight $\lambda$.

*Table 7.* Last performance on 10-task benchmarks under standard random sampling and class-imbalance settings.

| Sampling Strategy | ImageNet-R | CIFAR-100 | CUB-200 |
|---|---|---|---|
| Random | $82.77_{\pm 0.10}$ | $92.13_{\pm 0.19}$ | $89.77_{\pm 0.16}$ |
| Class-Imbalance | $82.62_{\pm 0.19}$ | $92.09_{\pm 0.19}$ | $89.64_{\pm 0.42}$ |

## C.4. Analysis of the Sampling Strategy for the Proxy Set

In our experiments, we employed a simple random sampling approach because we found the performance of $E^2$-LoRA to be remarkably stable. Even in an extreme class-imbalance scenario—where only a single category is sampled to compute the principal components—the performance remains consistent. Table 7 compares the Last-Acc under standard random sampling versus this extreme imbalance setting.

the principal directions of input representations in InfLoRA, and the principal directions of output feature drift in our $E^2$-LoRA, as shown in Figure 6. Experiments are conducted on ImageNet-R, where we capture the energy of the QKV linear projection layers in all 12 layers across all 10 tasks.

The results indicate that the energy in the parameter column vector space is uniformly distributed, which hinders the creation of free dimensions for subsequent tasks via space reduction. Applying SVD to the input representation space yields more concentrated energy than the parameter space; however, since the input features combine pretrained and task-specific components, the space remains high-dimensional and energy is still relatively dispersed. In contrast, the task-specific output drift space identified by our method exhibits higher energy concentration, allowing parameter updates to be focused on a few dominant directions, thereby preserving core knowledge while providing interference-free, orthogonal capacity for subsequent tasks.

## C.3. Sensitivity Analysis of the Hyperparameter $\lambda$

We conduct a sensitivity analysis on the distillation loss weight $\lambda$, as shown in Figure 7. When $\lambda$ ranges from 0.1 to 0.5, the performance remains relatively stable. In the absence of the distillation loss ($\lambda = 0$), performance on the Cars-196 dataset degrades noticeably, while the other datasets exhibit only slight drops. In all experiments reported in the main text, we set $\lambda = 0.2$.

