# OpenReview forum: "Energy-Structured Low-Rank Adaptation for Continual Learning"
_ICML.cc/2026/Conference — ICML 2026 regular_

### Official Review · Reviewer_BHbn · 2026-02-27

**Soundness:** 3
**Presentation:** 4
**Significance:** 3
**Originality:** 2
**Overall Recommendation:** 5
**Confidence:** 4

**Summary:**

This manuscript worked on the problem of continual learning with pre-trained models (CLPTM). By analyzing the existing orthogonalization methods, the authors pointed out that they usually adopted the orthogonalization between parameters, or between the input representation of a certain parametric module and its parameters. In this work, the authors proposed to analyze the output feature drift introduced by a task-specific LoRA module. Based on the theoretical analysis and the empirical observation, the authors proposed Energy-Concentrated and Energy-Ordered Low-Rank Adaptation (E2-LoRA). By ordering and concentrating the knowledge into the leading ranks of SVD decomposition, it introduced a dynamic rank allocation strategy to achieve a better trade-off between stability and plasticity. Experiments on some common datasets were conducted to support the effectiveness and advantages of this proposed method.

**Compliance With Llm Reviewing Policy:**

Affirmed.

**Final Justification:**

With the rebuttal phrase, my concerns have been addressed, especially the questions regarding the memory and computational overhead.

Based on this, I decided to increase my score from 4 to 5 and continue to support the acceptance of this manuscript.

**Key Questions For Authors:**

1. In Sec 4.1, the authors mentioned that they set the energy retention ratio $\rho$ as 0.9999, which is unusually strict for a truncation problem like that. I wonder how sensitive it is when choosing different values for this retention ratio. I failed to find further discussions regarding this part, even in the supplementary materials.
2. How about the computational overhead compared to the previous methods, especially the similar methods like InfLoRA? It would be great if the authors could provide further theoretical and empirical evidence regarding this point.
3. About the proxy input feature sampling (Step 15) of Algorithm 1. I noticed that the authors discussed the robustness regarding the size of this proxy dataset and mentioned the observation of "16 samples are sufficient to obtain a stable, energy-ordered and energy-concentrated orthogonal basis." I have some questions regarding this sampling process: I am curious about whether we need to apply some constraints within this sampling problem, e.g., class-balanced sampling to guarantee at least one sample per class.

**Limitations:**

There is no obvious limitation.

**Strengths And Weaknesses:**

## Strengths
- The theoretical and empirical analyses are promising. They provided a clear view for the motivation of this work.
- The proposed method is simple yet effective. It seems that the proposed framework is easy to implement.
- The experimental part is complete and supportive. The ablation studies are clear for understanding the effectiveness of the proposed component. - Experiments across multiple benchmarks (e.g., CIFAR-100, CUB-200, ImageNet-R) demonstrate consistent improvements over state-of-the-art methods and robustness in long task sequences.
-  The presentation of this manuscript is clear and easy to understand.

## Weaknesses
- Although this manuscript proposed to build the orthogonalization relation on the output space, the technique novelty is still limited compared to some previous studies like InfLoRA.
- It seems that the design of this method needs to store the $B$ an $A$ matrices for each task individually. Compared with some methods that can continually merge the LoRA weights to reduce the storage, this method is constrained to this design and the parameters cannot be merged.
- There is no further analysis regarding the computation overhead, especially comparisons with some previous similar methods like InfLoRA.

---

> ### Author Rebuttal · Authors · 2026-03-31
>
> We thank the reviewer for the insightful and constructive comments. Below we address each concern in detail.
>
> >**Improvements over InfLoRA**
>
> Compared to InfLoRA, our method introduces several key improvements:
> 1.  **Data-Driven Principal Directions.** We leverage the data distribution to estimate the principal components of the task-specific output drift after training. This leads to a more **concentrated representation of task knowledge**, where the most informative directions are explicitly identified and preserved.
> 2.  **More Efficient Rank Utilization.** By concentrating task-relevant information into a compact subspace, our method can **retain more knowledge with fewer ranks**. This improves parameter efficiency and reduces redundancy compared to input-space orthogonalization.
> 3.  **Reduced Inter-Task Interference.** The compact and energy-structured representation helps **better separate tasks in the output drift space**, thereby mitigating interference between tasks and improving continual learning performance.
> 4.  **Energy-Aware Dynamic Rank Allocation.** Building on this formulation, we propose an energy-aware dynamic rank allocation strategy, which **adaptively balances knowledge retention and model plasticity** by preserving high-energy components while reallocating low-energy dimensions.
> 5.  **Consistent Empirical Gains.** Our ablation studies show that under all evaluated settings, our method consistently outperforms InfLoRA. This validates the advantage of operating in the **output drift space** rather than the input space.
>
> >**Storage Analysis of $B$ and $A$ Matrices**
>
> 1. **Training Phase:** The additional **storage overhead remains bounded** and does not grow linearly with the number of tasks. This is because, when a new task arrives, the low-energy tail ranks of previous tasks are pruned and discarded, preventing accumulation over time. Specifically, the overhead per parameter matrix is approximately $d_{out}\times (d_{in}+d_{out})$. In comparison, although InfLoRA can merge $B$ and $A$ into the base parameters, it still needs to store orthogonal basis vectors and the current task branch. Its storage overhead is approximately $d_{in}\times d_{in}+r\times (d_{in}+d_{out})$, where $r$ denotes the rank of the LoRA.
> 2. **Inference Phase:** At inference time, all LoRA parameters can be merged into the backbone weights, incurring **no additional computational or memory overhead.**
>
> Furthermore, since all stored parameters from previous tasks are frozen and only a small number of LoRA parameters for the current task are updated, the overall GPU memory and computational cost remain modest. In practice, the memory footprint is comparable to that of InfLoRA, as shown below.
>
> >**Empirical Analysis of Computational and Memory Overhead**
>
> We conduct continual learning experiments on an NVIDIA H100 GPU with a batch size of 64. The per-task training time and peak memory usage are summarized below.
>
> Method|Peak Memory|Time / Task (ImageNet-R)|Time / Task (CIFAR-100)
> -|-|-|-
> InfLoRA|18.3 GB|139 s|240 s
> Ours|16.9 GB|147 s|238 s
>
> The results show that our method has comparable per-epoch computational and memory overhead to InfLoRA.
>
> >**Sensitivity Analysis of the Energy Retention Ratio $\rho$**
>
> In our framework, continual learning is performed on top of a pre-trained model, where the output feature drift lies in a highly concentrated subspace. Empirically, we observe that retaining only a small fraction of ranks (on average, around 10%) is sufficient to preserve over 99.99% of the energy. Moreover, our method enforces a minimum rank allocation for each new task, which further stabilizes performance and mitigates potential under-allocation. As a result, **the model is relatively robust to the choice of $\rho$**. We report the Last-Acc under different settings of $\rho$ in the table below.
>
> $\rho$|ImageNet-R|CIFAR-100
> -|-|-
> 0.99|82.0|91.3
> 0.999|82.6|91.8
> 0.9999|82.8|92.1
> 0.99999|82.7|92.2
>
> >**Analysis of the Sampling Strategy for the Proxy Set**
>
> In our experiments, we employed a simple **random sampling** approach because we found the performance of $E^2$-LoRA to be remarkably stable. Even in an extreme **class-imbalance** scenario—where only a single category is sampled to compute the principal components—the performance remains consistent. The table below compares the Last-Acc under standard random sampling versus this extreme imbalance setting.
>
> Sampling Strategy|ImageNet-R|CIFAR-100|CUB-200
> -|-|-|-
> Random|$82.77\pm0.10$|$92.13\pm0.19$|$89.77\pm0.16$
> Class Imbalance|$82.62\pm0.19$|$92.09\pm0.19$|$89.64\pm0.42$

---

> > ### Author Rebuttal · Reviewer_BHbn · 2026-04-03
> >
> > Thanks to the authors for providing further explanations to my questions or concerns. After this phrase, I think my concerns have been addressed, especially the questions regarding the memory and computational overhead.
> >
> > Based on this, I decided to increase my score from 4 to 5 and continue to support the acceptance of this manuscript.

---

### Official Review · Reviewer_hXWb · 2026-03-08

**Soundness:** 3
**Presentation:** 3
**Significance:** 2
**Originality:** 2
**Overall Recommendation:** 4
**Confidence:** 4

**Summary:**

This paper proposes E^2-LoRA, which addresses the stability-plasticity dilemma by leveraging the spectral properties of task-specific output feature drift. The core insight is that the output feature drift induced by LoRA fine-tuning resides in a low-dimensional, energy-concentrated subspace. The authors theoretically prove that preserving parameters along the principal directions of this drift minimizes output reconstruction error under rank constraints. Building on this, E²-LoRA introduces an energy-structured transformation, a dynamic rank allocation strategy, and output feature drift-induced orthogonalization. Extensive experiments across class-incremental and domain-incremental benchmarks demonstrate state-of-the-art performance.

**Compliance With Llm Reviewing Policy:**

Affirmed.

**Final Justification:**

During the rebuttal phase, the authors addressed most of my concerns. Although the clarity in some parts of the paper could still be improved, I am inclined to recommend acceptance of the paper.

**Key Questions For Authors:**

1. The use of self-distillation and classifier alignment appears to have no direct relationship with the core proposed method. Could the authors justify why this particular combination is uniquely effective?

2. The method is evaluated with LoRA modules in ViT backbones. How might it extend to other architectures (e.g., vision-language models)?

**Limitations:**

yes

**Strengths And Weaknesses:**

Strengths:

1. Experiments across multiple datasets and continual learning settings effectively validate the effectiveness of the proposed method.

2. The related work section thoroughly positions the method against parameter-space (O-LoRA) and input-space (InfLoRA), clearly articulating the novel output-drift perspective.

3. The paper is well-written and easy to follow. The visualizations effectively contrast orthogonalization strategies and illustrate the pipeline of the proposed method.

Weaknesses:

1. The connection between the theoretical error bounds (Propositions 3.1–3.2) and the practical dynamic rank allocation strategy (Section 3.3) could be made more explicit.

2. Existing research has explored designing specialized LoRA algorithms for continual learning. This paper lacks a comparison with such methods (e.g., CL-LoRA [1]).

3. Several components (self-distillation, classifier alignment, proxy-based SVD) are adapted from prior work. The significance of the ablation study regarding these components (Table 4) is not entirely clear.

[1] CL-LoRA: Continual Low-Rank Adaptation for Rehearsal-Free
Class-Incremental Learning

---

> ### Author Rebuttal · Authors · 2026-03-31
>
> We thank the reviewer for the insightful and constructive comments. Below we address each concern in detail.
>
> >**Connection Between Theoretical Error Bounds (Propositions 3.1–3.2) and Dynamic Rank Allocation (Section 3.3)**
>
> 1. **Proposition 3.1** establishes the optimality of rank truncation in the output drift space. It guarantees that, during dynamic rank allocation, truncating the ranks of previous tasks when allocating orthogonal subspaces for new tasks leads to the **minimal impact on the outputs of prior tasks**.
> 2. **Proposition 3.2** further provides the corresponding error bound. It derives the **expected output error introduced by truncating the ranks of previous tasks** within the dynamic rank allocation process.
>
> This mechanism ensures that when we compress the subspaces of past tasks to free up orthogonal capacity for new tasks, the impact on previously learned knowledge is both **mathematically bounded and minimized**. As a result, our approach offers a principled way to balance **model stability** (by preserving high-energy components of prior tasks) and **plasticity** (by reallocating low-energy, less informative dimensions to accommodate new tasks).
>
> >**Comparison with CL-LoRA**
>
> We conducted additional experiments to compare our method with CL-LoRA across multiple benchmarks. The results, summarized in the table below, demonstrate the consistent superiority of our approach in terms of Last-Acc / Inc-Acc.
> Method|ImageNet-R|CIFAR-100|ImageNet-A
> -|-|-|-
> CL-LoRA [1]|80.3 / 86.3|87.1 / 92.1|60.5 / 70.2
> Ours|82.8 / 87.2|92.1 / 95.0|64.3 / 71.5
>
> >**On the Role of Self-Distillation and Classifier Alignment**
>
> We thank the reviewer for this important question. We would like to clarify that **self-distillation (KD) and classifier alignment (CA) are not core to our method**, but rather standard auxiliary components commonly used in continual learning.
>
> 1. **Not a source of novelty or performance gain.** KD and CA are widely adopted to alleviate forgetting and classifier bias, and are used in many prior works. Our contribution does not rely on these modules, nor do we claim any novelty from them.
> 2. **Isolating the effect of our method.** The key purpose of **Table 4** is to explicitly **isolate the contribution of our output-drift-based orthogonalization** from these auxiliary components. We evaluate all combinations, including **removing both KD and CA entirely**, and observe that our method **consistently outperforms parameter-space and input-space orthogonalization in all cases**. This demonstrates that the performance gains are **independent of KD and CA**.
>
> This confirms that the effectiveness of our method stems from the proposed energy-structured output space formulation, rather than any specific combination with KD or CA.
>
> >**Extension to Vision-Language Models (VLMs)**
>
> In the main paper, we evaluate our method on standard CIL and DIL benchmarks using ViT backbones. To address the reviewer’s question regarding generalization to other architectures, we further extend our method to **vision-language models (VLMs)**. Specifically, we conduct continual learning experiments based on a pre-trained CLIP model. For fair comparison, all methods are initialized from the same pre-trained CLIP and trained **without exemplars**. The results show that our method **consistently outperforms competing approaches** in this setting, demonstrating that our approach generalizes effectively beyond pure vision backbones and can be naturally applied to VLM architectures.
>
> Method|ImageNet-R|CUB-200|ImageNet-A
> -|-|-|-
> RAPF [2]|79.62 / 86.28|76.34 / 83.04|55.37 / 67.32
> PROOF [3]|80.10/ 85.34|79.43 / 84.93|55.67 / 65.50
> SimpleCIL [4]|74.48 / 81.06|77.52 / 83.81|-
> ENGINE [5]|80.37 / 86.22|80.20 / 86.65|-
> SECA [6]|83.18 / 88.58|-|65.09 / 74.45
> Ours|84.02 / 88.59|87.53 / 91.97|65.79 / 75.19
>
> [1] CL-LoRA: Continual Low-Rank Adaptation for Rehearsal-Free Class-Incremental Learning. CVPR 2025.
>
> [2] Class-Incremental Learning with CLIP: Adaptive Representation Adjustment and Parameter Fusion. ECCV 2024.
>
> [3] Learning Without Forgetting for Vision-Language Models. TPAMI 2025.
>
> [4] Revisiting Class-Incremental Learning with Pre-Trained Models: Generalizability and Adaptivity are All You Need. IJCV 2025.
>
> [5] External Knowledge Injection for CLIP-Based Class-Incremental Learning. ICCV 2025.
>
> [6] Harnessing Textual Semantic Priors for Knowledge Transfer and Refinement in CLIP-Driven Continual Learning. AAAI 2026.

---

> > ### Author Rebuttal · Reviewer_hXWb · 2026-04-03
> >
> > I thank the authors for their comprehensive response. Most of my concerns have been addressed, and I encourage the authors to make the connection between the theoretical error bounds (Propositions 3.1–3.2) and the practical dynamic rank allocation strategy (Section 3.3) more explicit in the main text. I will raise my score.

---

### Official Review · Reviewer_SA72 · 2026-03-10

**Soundness:** 4
**Presentation:** 4
**Significance:** 3
**Originality:** 4
**Overall Recommendation:** 5
**Confidence:** 5

**Summary:**

This paper addresses the challenge of catastrophic forgetting in incremental learning scenarios. The main contribution is a data-driven optimization subspace management technique that enables the effective separation of LoRA adapters for sequentially arriving tasks. This separation is achieved via principal component projection of the current adapters' weights, guided by shifts in the output distribution. The authors demonstrate that their method outperforms existing state-of-the-art approaches and achieves performance comparable to full-task fine-tuning.

**Compliance With Llm Reviewing Policy:**

Affirmed.

**Key Questions For Authors:**

1. How powerfully does the choice of &\rho& impacts on the performance of the model? Could you provide additional ablation on its connection with models' stability and plasticity?

2. Could you support your paper with more rich architectural exploration?  May smaller networks demand greater proxy datasets to extract the principal components more carefully to avoid inefficient rank usage in case of &d_{out}& restraint?

3. E^2-LoRA's energy-based ordering assumes tasks can be separated orthogonally. But if similar classes (with close representation in mean of cosine distance, for instance) appear in different tasks, wouldn't the model need to 're-learn' similar features using new basis, duplicating representations and wasting rank capacity? Could you analyze performance under different task orderings, especially when semantically related classes are split?"

**Limitations:**

The existing approach need to be tested on NLP tasks to prove its generality. More investigation on the nature of proxy dataset size and sampling is requested.

**Strengths And Weaknesses:**

Strengths
1. State-of-the-art Performance: The survey provide a comprehensive evaluation over class- and domain-incremental benchmarks such as ImageNet-R, CIFAR-100, CUB-200, Cars-196 and Office-Home, DomainNet. Performance is compared to recent baselines, such as L2P, DualPrompt, SLCA, TUNA, MOS, etc and demonstrates consistent dominance of E^2-LoRA both in final accuracy and aggregated accuracy dynamics. The final quality approaches to that of a joint tuning.
2. Strong Theoretical Foundation: The authors suggested the rigorous proof of optimality guaranteeing the proposed PCA-based task alignment minimizes the error induced by rank freeing procedure. This theoretical grounding adds significant credibility to the approach.
3.Simplified Subspace Management: E^2-LoRA introduces a novel mechanism for managing optimization subspaces by leveraging output drift transformation. According to the authors, the PCA-based projection estimation requires only a minimal number of samples and can be performed after a training stage, thereby avoiding any interference with previously learned tasks. Therefore, E^2-LoRA offers an efficient data-driven solution for continual learning scenarios.

Weaknesses

1. Limited Model Scope: The evaluation is restricted to a single visual transformer architecture, leaving the generalizability of the approach to other model families. Moreover, the effectiveness of the proposed mechanism is not estimated comprehensively on different $d_{out}$ values.


2. Insufficient Proxy Dataset Analysis: The experiments are confined to a narrow set of benchmarks, and the paper fails to address class-specific challenges, particularly when representations of new classes closely resemble those of old ones. Additionally, no evidence is provided to support the method's effectiveness in a domain-incremental learning setup.


3. Unexplored Hyperparameter Sensitivity: Although dynamic rank allocation is presented as a core contribution, the critical hyperparameter $\rho$ controlling rank rearrangement is not adequately explored. The paper does not analyze the sensitivity of the model to this parameter, nor does it discuss the trade-offs involved in its selection. Notably, low values of $\rho$ risk exacerbating forgetting, while high values may unnecessarily restrict the freeing of rank capacity, thereby limiting model plasticity.

---

> ### Author Rebuttal · Authors · 2026-03-31
>
> We thank the reviewer for the insightful and constructive comments. Below we address each concern in detail.
>
> >**Effect of Proxy Dataset Size on Smaller Models (Low $d_{out}$)**
>
> The proxy-based principal component estimation is highly robust. **Even for smaller models with limited dimensions, the required number of samples remains very low.** We experiment on smaller models, ViT-Small and ViT-Tiny, varying the number of proxy samples $D$ and reporting Last-Acc on the ImageNet-R 10-task benchmark as below.
> Method|ViT-Small|ViT-Tiny
> -|-|-
> MOS [1]|71.27|57.60
> TUNA [2]|71.75|56.18
> Ours ($D=4$)|75.05|62.35
> Ours ($D=8$)|75.12|62.40
> Ours ($D=16$)|75.12|62.37
> Ours ($D=32$)|75.13|62.43
>
> >**Sensitivity Analysis of the Energy Retention Ratio $\rho$**
>
> Due to the highly concentrated output drift space, preserving only about 10% of the ranks retains over 99.99% of the energy. Our method also allocates a minimum rank per task, which stabilizes performance and prevents under-allocation. As a result, **the model is relatively robust to the choice of $\rho$**. We report the Last-Acc under different settings of $\rho$ in the table below.
> $\rho$|ImageNet-R|CIFAR-100
> -|-|-
> 0.99|82.0|91.3
> 0.999|82.6|91.8
> 0.9999|82.8|92.1
> 0.99999|82.7|92.2
>
> >**Generalization to Other Architectures**
>
> In the main paper, we evaluate our method on standard CIL and DIL benchmarks using ViT backbones. To address the reviewer’s question regarding generalization to other architectures, we extend our method to CLIP. The Last-Acc in the table shows that our method consistently achieves the best performance, demonstrating strong generalization to VLMs.
> Method|ImageNet-R|CUB-200|ImageNet-A
> -|-|-|-
> RAPF [3]|79.6|76.3|55.4
> PROOF [4]|80.1|79.4|55.7
> ENGINE [5]|80.4|80.2|-
> SECA [6]|83.2|-|65.1
> Ours|84.0|87.5|65.8
>
> >**Analysis of the Sampling Strategy for the Proxy Set**
>
> In our experiments, we employed a simple **random sampling** approach because we found the performance of $E^2$-LoRA to be remarkably stable. Even in an extreme **class-imbalance** scenario—where only a single category is sampled to compute the principal components—the performance remains consistent. The table below compares the Last-Acc under standard random sampling versus this extreme imbalance setting.
> Sampling Strategy|ImageNet-R|CIFAR-100|CUB-200
> -|-|-|-
> Random|$82.77\pm0.10$|$92.13\pm0.19$|$89.77\pm0.16$
> Class Imbalance|$82.62\pm0.19$|$92.09\pm0.19$|$89.64\pm0.42$
>
> >**Effectiveness in the Domain-Incremental Learning Setup**
>
> The effectiveness of our method in the DIL setup is empirically validated in **Table 3** of the manuscript. The theoretical foundation of our success in DIL lies in the energy-structured transformation within the output drift space. By isolating domain-specific updates into energy-ordered orthogonal subspaces, $E^2$-LoRA effectively mitigates catastrophic forgetting of previous domains while preserving the model's plasticity to adapt to new input distributions.
>
> >**Special Case: Semantically Related Classes Split Across Tasks**
>
> 1. **No Redundant Rank Usage for Similar Classes.** New tasks build on previously adapted parameters, reusing shared knowledge and only learning low-rank residuals. Thus, similar classes do not cause duplicated representations.
> 2. **Analysis under Different Task Orderings.** In our main paper, we follow standard benchmarks with **random class ordering (Random)**. To address the reviewer’s concern, we construct a more challenging setting where **semantically similar classes are intentionally split across different tasks (Semantic Split)**. We report the Last-Acc under both settings below:
> Method (Task Ordering)|ImageNet-R|CIFAR-100
> -|-|-
> MOS [1] (Random)|$78.10\pm0.12$|$90.53\pm0.35$
> MOS [1] (Semantic Split)|$77.67\pm0.38$|$90.30\pm0.29$|
> TUNA [2] (Random)|$79.44\pm0.38$|$91.79\pm0.15$
> TUNA [2] (Semantic Split)|$79.17\pm0.29$|$91.72\pm0.23$
> Ours (Random)|$82.77\pm0.10$|$92.13\pm0.19$
> Ours (Semantic Split)|$82.24\pm0.34$|$92.06\pm0.12$
>
> The results show that separating semantically similar classes into different tasks introduces a slightly more challenging scenario, as these classes are no longer jointly optimized for discrimination within the same task. This leads to a minor performance drop across all methods. Our method achieves the best performance under both task orderings, showing effective knowledge reuse without inefficient rank usage.
>
> [1] MOS: Model Surgery for Pre-Trained Model-Based Class-Incremental Learning. AAAI 2025.
>
> [2] Integrating Task-Specific and Universal Adapters for  Pre-Trained Model-based Class-Incremental Learning. ICCV 2025.
>
> [3] Class-Incremental Learning with CLIP: Adaptive Representation Adjustment and Parameter Fusion. ECCV 2024.
>
> [4] Learning Without Forgetting for Vision-Language Models. TPAMI 2025.
>
> [5] External Knowledge Injection for CLIP-Based Class-Incremental Learning. ICCV 2025.
>
> [6] Harnessing Textual Semantic Priors for Knowledge Transfer and Refinement in CLIP-Driven Continual Learning. AAAI 2026.

---

> > ### Author Rebuttal · Reviewer_SA72 · 2026-04-03
> >
> > Thank authors for the reply. The authors have addressed most of my concerns.

---

### Decision · Program_Chairs · 2026-04-30

**Decision:**

Accept (regular)

**Comment:**

I recommend acceptance.

Reviewers found the paper technically sound and supported by solid empirical evidence. The paper’s main contribution is to rethink orthogonalization-based continual LoRA through output feature drift, and to use this formulation to support rank concentration and reuse for continual adaptation. The empirical results are broad and provide convincing support for the main claims.

Most concerns were addressed in the rebuttal, and the remaining weaknesses are limited. The authors are encouraged to incorporate the key clarifications from the rebuttal into the final version of the paper.